# Host-microbe co-metabolism via MCAD generates circulating metabolites including hippuric acid

Kali M. Pruss[1], Haoqing Chen[2], Yuanyuan Liu[2], William Van Treuren[1],
Steven K. Higginbottom[1], John B. Jarman[2], Curt R. Fischer[3,8], Justin Mak[4],
Beverly Wong[4], Tina M. Cowan[2], Michael A. Fischbach [1,3,5,6],
Justin L. Sonnenburg [1,6,7] & Dylan Dodd [1,2] ✉

The human gut microbiota produces dozens of small molecules that circulate in blood, accumulate to comparable levels as pharmaceutical drugs, and influence host physiology. Despite the importance of these metabolites to human health and disease, the origin of most microbially-produced molecules and their fate in the host remains largely unknown. Here, we uncover a host-microbe co-metabolic pathway for generation of hippuric acid, one of the most abundant organic acids in mammalian urine. Combining stable isotope tracing with bacterial and host genetics, we demonstrate reduction of phenylalanine to phenylpropionic acid by gut bacteria; the host re-oxidizes phenylpropionic acid involving medium-chain acyl-CoA dehydrogenase (MCAD). Generation of germ-free male and female MCAD$^{-/-}$ mice enabled gnotobiotic colonization combined with untargeted metabolomics to identify additional microbial metabolites processed by MCAD in host circulation. Our findings uncover a host-microbe pathway for the abundant, non-toxic phenylalanine metabolite hippurate and identify β-oxidation via MCAD as a novel mechanism by which mammals metabolize microbiota-derived metabolites.

The human gut microbiome produces numerous drug-like small molecules that impact various aspects of human biology[1–4]. These molecules are produced in the gut via metabolism of diet- and host-derived molecules such as glycans, proteins, and amino acids. Microbiota-dependent metabolites impact physiology locally in the gut where their concentrations are the highest. However, a subset are absorbed across the intestinal wall and circulate throughout the body where they may influence host physiology at organs distal to the gut[5–9]. Despite extensive recent literature documenting the impact of microbial metabolites on the host, significant gaps exist in our knowledge about these molecules. Chief among these are the lack of clarity on how microbiota-generated metabolites intersect

with host metabolism, and the identity and fate of the major products. Such insight is essential to understand how host genetics influence the spectrum of microbial metabolites within an individual and will identify novel biomarkers for microbial functions in the gut.

Gut microbial metabolites enter host circulation where they are metabolized by enzymes involved in phase I (modification via oxidation, reduction, and hydrolysis) and phase II metabolism (conjugation with glutathione, sulfate, glucuronic acid, or amino acids—typically glycine or glutamine), classically known for their roles in drug metabolism. Products of phase I and phase II metabolism of microbial metabolites include: (1) indoxyl sulfate[10–12], a host-microbe

[1]Department of Microbiology and Immunology, Stanford University School of Medicine, Stanford, CA, USA. [2]Department of Pathology Stanford University School of Medicine, Stanford, CA, USA. [3]ChEM-H, Stanford University, Stanford, CA, USA. [4]Stanford Healthcare, Palo Alto, CA, USA. [5]Department of Bioengineering, Stanford University, Stanford, CA, USA. [6]Chan Zuckerberg Biohub, San Francisco, CA, USA. [7]Center for Human Microbiome Studies, Stanford, CA, USA. [8]Present address: Octant Bio, Emeryville, CA, USA. ✉e-mail: ddodd2@stanford.edu

co-metabolite of tryptophan whose plasma levels rise as the kidneys fail, and is thought to contribute to the cardiovascular sequelae in end-stage kidney disease[13, 14]; (2) *p*-cresol sulfate, the sulfated product of microbial tyrosine degradation, is associated with cardiovascular and kidney damage[9, 15]; (3) trimethylamine *N*-oxide, a host-microbe co-metabolite of trimethyl-containing compounds such as choline, whose plasma levels are associated with adverse outcomes in patients with cardiovascular disease[7]. Many colon-derived metabolites in humans are subjected to sulfation and glucuronidation reactions[16], but the extent to which alternative host metabolic pathways contribute to the large remainder of microbially-derived metabolites remains to be defined.

We previously demonstrated the role of the phenyllactate dehydratase (FldABC, Fig. 1a) gene cluster in reductive metabolism

of aromatic amino acids by gut bacteria, accounting for nine metabolites that circulate in the host[17], including indolepropionic acid (IPA)−a microbial metabolite of tryptophan that modulates intestinal permeability[17, 18]. Here, we sought to understand how the host metabolizes additional compounds generated by the *fld* locus that enter circulation (Supplementary Fig. 1). Using metabolic profiling and gnotobiotic colonization of wild-type (*wt*) mice, we find that the *fld* locus is a key determinant for one of the most concentrated human urine metabolites, hippuric acid. Gnotobiotic experiments with transgenic mice reveal that products of the *fld* locus are subjected to β-oxidation in the host via medium chain acyl-CoA dehydrogenase (MCAD). Our findings uncover a previously unappreciated mechanism for host-microbe co-metabolism that contributes to metabolites that circulate in the host.

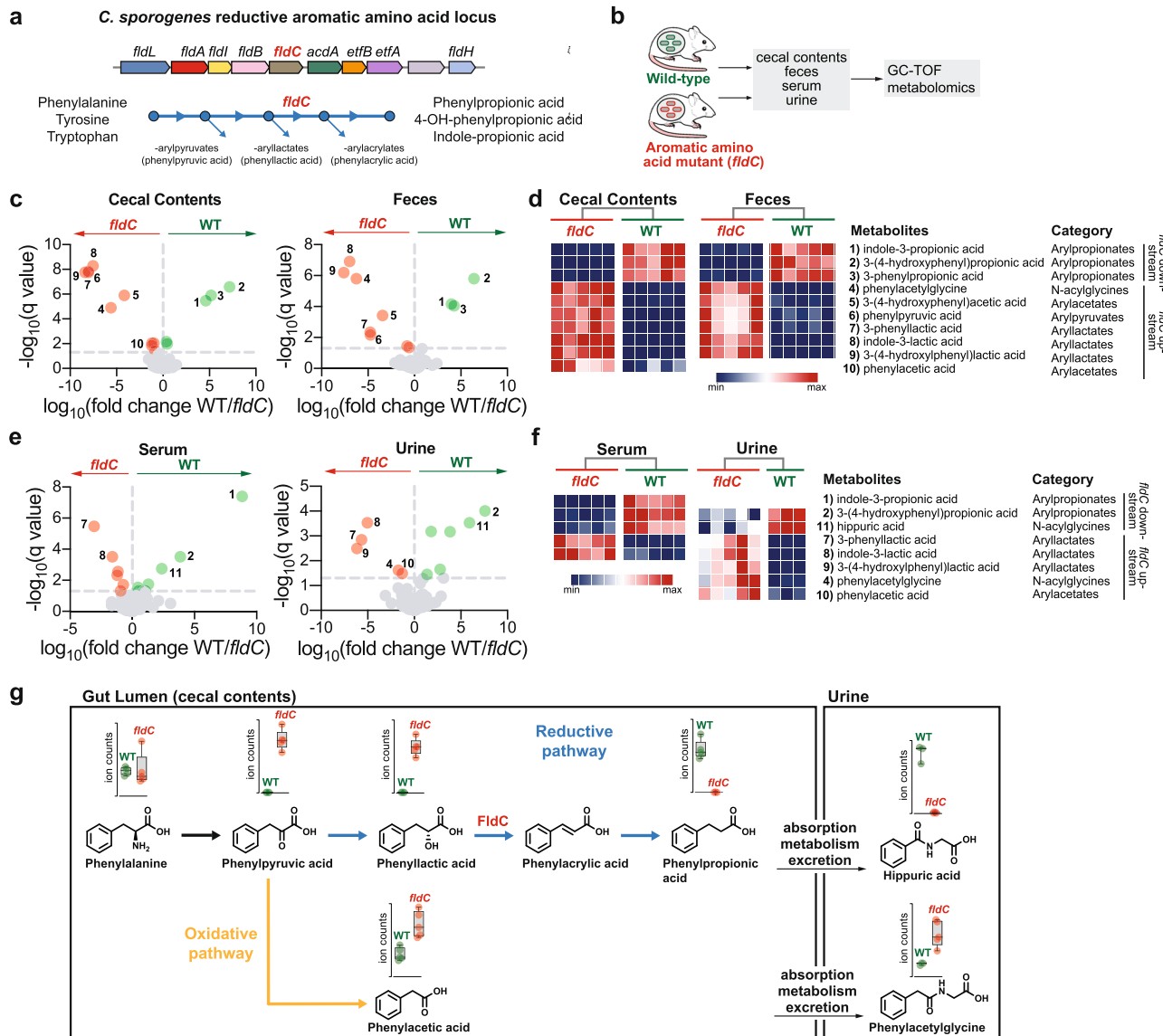

**Fig. 1 | A gut bacterial gene locus influences the levels of numerous metabolites that circulate in the host. a** Schematic overview of the *C. sporogenes* gene locus and encoded biochemical pathway responsible for reductive aromatic amino acid metabolism. *fldC*, phenyllactate dehydratase subunit C. **b** Germ-free Swiss Webster mice were mono-colonized with either wild-type *C. sporogenes* or its *fldC* mutant. Body fluids were collected and analyzed by GC-TOF shotgun metabolomics. **c, e** Metabolites significantly enriched in WT (green, positive fold-change) or *fldC*-colonized mice (red, negative); significant metabolites are colored, paired two-tailed *t*-tests with two-stage linear step-up procedure of Benjamini, Krieger, and Yekutieli, adjusted *P*-value < 0.05. **d, f** Metabolites corresponding to (**c, e**) that differ significantly between WT and *fldC*-colonized mice; each column represents row-normalized peak intensity for an individual mouse, metabolites were included in the heatmap if significant in at least two host compartments sampled. All significant discoveries are reported in Supplementary Data 2. **g** *fldC*-dependent alterations in levels of metabolites in the oxidative and reductive pathways for phenylalanine metabolism in the gut lumen and potentially linked metabolites in host urine. The mouse schematic in **b** was adapted from ref. [17].

## Results

### Shotgun metabolomics identifies *fldC*-dependent metabolites in host circulation

Germ-free mice were mono-colonized with either wild-type (WT) *Clostridium sporogenes* or its phenyllactate dehydratase mutant (*fldC*) and cecal contents, feces, serum, and urine were collected for metabolomic profiling by gas chromatography time-of-flight mass spectrometry (GC-TOF-MS) (Fig. 1b, Supplementary Data 1, 2). Consistent with the role of the *fldC* gene in reductive aromatic amino acid metabolism, in feces and cecal contents of *fldC*-colonized mice we detected (i) a loss of arylpropionate end products, (ii) a build-up of intermediates proximal to the block in the pathway (aryllactates and arylpyruvates), and (iii) an increase in oxidative pathway products (arylacetates) (Fig. 1c, d). Similarly, in the serum and urine of *fldC*-colonized mice we observed diminished levels of arylpropionates, increased aryllactates, and an increase in the host-modified oxidative pathway product, phenylacetylglycine (compound 4, a host glycine conjugate of phenylacetate) (Fig. 1e, f). These results confirm our previous findings that a single microbial gene locus determines the abundance of several aromatic amino acid metabolites within the gut lumen and host circulation[17].

Intriguingly, our metabolomics analysis identified hippuric acid as one of the most highly enriched metabolites in serum and urine of WT vs. *fldC*-colonized mice (Fig. 1e, f, compound 11). The identification of hippuric acid is notable for the following three reasons: (1) hippuric acid has long been known to be one of the most abundant organic acids in mammalian urine[19], (2) the gut microbiota has been shown to contribute to hippuric acid in the host[4,20,21], yet no microbe or pathway has been shown to influence its levels, (3) while dietary benzoate[22] and toluene[23] are well known precursors, and other dietary compounds including phenylalanine[24] and polyphenols[19] have been associated with increased levels of hippuric acid, the contribution of microbial amino acid metabolism in the gut to host hippuric acid has not been previously demonstrated. Furthermore, abundance of the end product of oxidative phenylalanine metabolism, the cardiotoxic molecule phenylacetylglycine[8], was higher in the urine of *fldC*- colonized mice, indicating that reductive microbial metabolism of phenylalanine may represent a healthy alternative. Given that phenylpropionic acid was abundant in the feces and cecal contents of WT-colonized mice (Fig. 1c, d; compound 3), yet undetectable in serum and urine, we reasoned that microbially-produced phenylpropionic acid could be converted to hippuric acid by the host (Fig. 1g).

### *C. sporogenes* reduction of phenylalanine to phenylpropionic acid yields hippuric acid

To further explore the role of the *fldC* gene cluster in hippuric acid production, we first validated our GC-TOF-MS findings by colonizing mice with WT *C. sporogenes* or the *fldC* mutant and analyzing hippuric acid levels by stable isotope dilution liquid chromatography–mass spectrometry (LC-MS). These experiments revealed that urine hippuric acid levels in *fldC*- colonized mice were low—similar to germ-free mice — whereas WT-colonized mice had millimolar levels of urine hippuric acid (Fig. 2a). We next asked whether orally administered phenylpropionic acid could serve as a precursor for urine hippuric acid. We orally gavaged germ-free mice with stable isotope-labeled phenylpropionic acid ($d_9$-phenylpropionic acid) and monitored label incorporation into hippuric acid ($d_5$-hippuric acid) (Fig. 2b). Labeled hippuric acid was undetectable at the time of $d_9$-phenylpropionic acid gavage ($t = 0$); however, we noted a rise in labeled hippuric acid at the 3 and 6 h timepoints followed by a drop at 9 h, and levels were undetectable by 24 h post-gavage suggesting that phenylpropionic acid is converted to hippuric acid by the host (Fig. 2c). To directly test the hypothesis that hippuric acid arises from phenylalanine metabolism involving the *fldC* gene, we performed a similar stable isotope tracing experiment in germ-free and colonized mice, this time using labeled phenylalanine

($d_5$-phenylalanine). In germ-free mice, no labeled hippuric acid was detected (Fig. 2e), suggesting no endogenous pathway exists for hippuric acid production from phenylalanine. However, we noted high levels of labeled hippuric acid in the urine of WT-colonized mice, whereas the levels were undetectable in *fldC*-colonized mice (Fig. 2e). Taken together, these results reveal a new host-microbe co-metabolic pathway for conversion of dietary phenylalanine to hippuric acid requiring the *fld* locus.

### Hippuric acid and PPA levels vary in conventional mice from different vendors

Having identified a host-microbe co-metabolic pathway for hippurate production in mono-colonized gnotobiotic mice, we next asked whether these compounds are present in mice colonized with a conventional microbiota. To address this, we obtained isogenic Swiss Webster mice that were conventionally-raised in three different animal facilities (Jackson labs, Taconic, Charles River, Fig. 3a). We then quantified hippuric acid and phenylpropionic acid in the plasma and urine of these mice by LC-MS. Intriguingly, we found that hippuric acid levels were present in the plasma at micromolar levels only in mice raised at Jackson Labs (Jackson) (Fig. 3b). Consistent with the role of phenylpropionic acid as a precursor for hippuric acid, its levels were also significantly higher in the plasma of conventionally raised mice from Jackson Labs (Fig. 3c). Hippuric acid was similarly elevated in the urine of mice raised at Jackson labs (Fig. 3d). To confirm that the gut microbiota was responsible for the high levels of these compounds, we obtained cecal contents from mice from Jackson and Charles Rivers labs and transplanted them to germ-free Swiss Webster mice (Fig. 3e). These experiments revealed that only the cecal contents obtained from mice raised at Jackson labs conferred the phenotype of high levels of hippuric acid and phenylpropionic acid in plasma (Fig. 3f, g) and hippurate in urine (Fig. 3h) of ex germ-free mice. These results highlight two key findings: First, that hippuric acid and its precursor phenylpropionic acid circulate in the blood of conventional mice at micromolar levels. Second, there is substantial variability in hippuric acid levels across mice raised at different facilities, suggesting that variations in the microbiota contribute to changes in the circulating metabolome.

### Host MCAD generates hippuric acid from microbially-produced phenylpropionic acid

Having determined the microbial pathway contributing to hippuric acid generation, we sought to understand how the host produces hippuric acid from phenylpropionic acid. The oxidative pathway product of phenylalanine (phenylacetate) is conjugated with glycine by the host, yielding phenylacetylglycine, as seen in *fldC*-colonized mice (Fig. 1f). However, we failed to detect the corresponding glycine conjugate of the reductive pathway product of phenylalanine, phenylpropionylglycine. We hypothesized that phenylpropionic acid may be chain-shortened by the host via β-oxidation yielding benzoic acid and then conjugated with glycine to form hippuric acid (also known as benzoylglycine). To test this hypothesis, we re-derived medium chain acyl-CoA dehydrogenase (MCAD) knockout mice (MCAD$^{-/-}$)[25] as germ-free to enable gnotobiotic colonization experiments. Consistent with previous studies of conventionally-raised MCAD$^{-/-}$ mice, we noted a distinctive biochemical defect in GF MCAD$^{-/-}$ mice characterized by higher levels of serum acylcarnitines (C6, C8, C10:1, and C10) (Supplementary Fig. 2a) and of urinary suberic acid and hexanoylglycine (Supplementary Fig. 2b) after a 24-h fasting period.

To test whether MCAD is involved in hippuric acid production, we assayed hippuric acid levels in the urine of germ-free and WT *C. sporogenes*-colonized MCAD$^{-/-}$ and MCAD$^{+/+}$ mice. These experiments revealed that *C. sporogenes*-colonized MCAD$^{-/-}$ mice have reduced levels of urine hippuric acid compared to MCAD$^{+/+}$ mice (Fig. 4a). In addition, we found that MCAD$^{-/-}$ mice accumulate

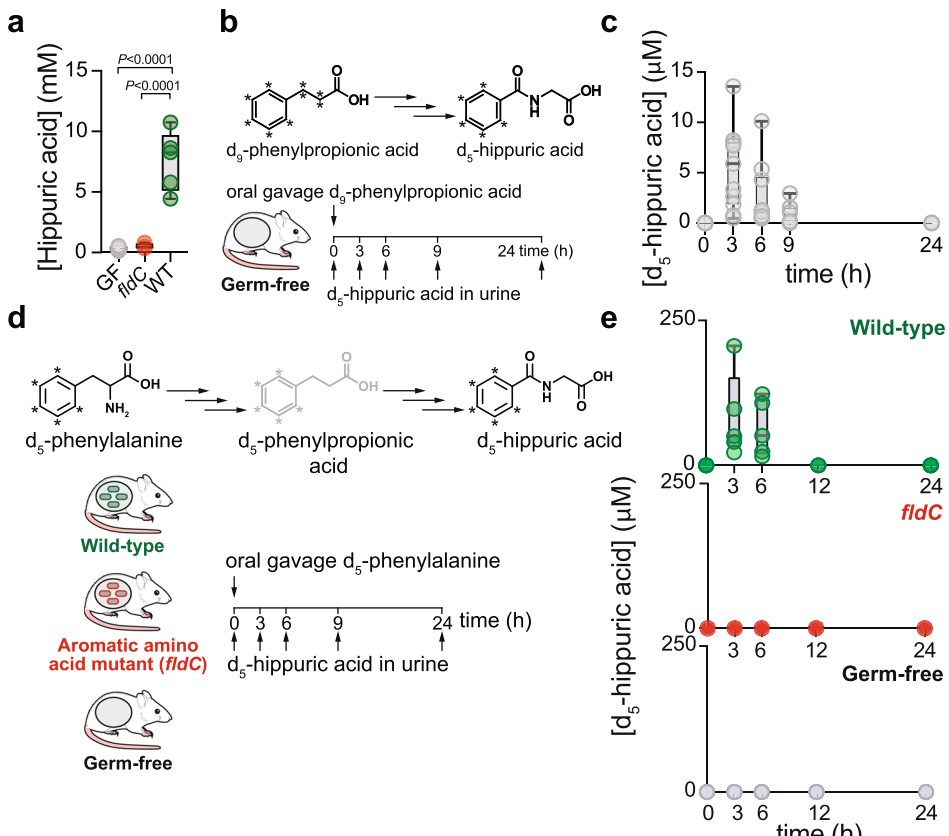

**Fig. 2 | Stable isotope tracing to map the conversion of phenylalanine to hippuric acid involving the *fldC* locus. a** Urine was collected from germ-free mice ($n = 10$), *fldC*-colonized mice ($n = 4$), or wild-type *C. sporogenes*-colonized mice ($n = 5$) and hippuric acid was measured using stable isotope dilution mass spectrometry (One-way ANOVA with Tukey's post-hoc multiple comparisons, $F_{(2, 16)} = 60.90$). **b** Germ-free mice were gavaged with $d_9$-phenylpropionic acid and incorporation of ring label into hippuric acid was quantified to determine whether phenylpropionic acid is a precursor to hippuric acid. **c** Administration of $d_9$-phenylpropionic acid to germ-free mice leads to accumulation of $d_5$-hippuric acid in urine ($n = 12$ mice). **d** Germ-free ($n = 10$ mice), *fldC*-colonized ($n = 5$), or WT-colonized ($n = 5$) mice were orally administered isotopically labeled phenylalanine and labeled hippuric acid was monitored in the urine over time. **e** Urine $d_5$-hippuric acid measured 0, 3, 6, 9, or 24 h following oral administration of $d_5$-phenylalanine using stable isotope dilution mass spectrometry. For **a**, **c**, **e**: boxes denote the median with inter-quartile distance; whiskers, maxima and minima. In (**c**), one outlier at $t = 3$ h (55.88 μM $d_5$-hippuric acid) was tested for and removed with the Extreme Studentized Deviate method implemented in GraphPad Prism 8. The mouse schematic in **b**, **d** was adapted from ref. [17].

phenylpropionylglycine in urine at levels complementary to the defect in hippuric acid concentrations (Fig. 4b). These findings suggest that in MCAD$^{-/-}$ mice, β-oxidation of phenylpropionic acid is partially blocked, prompting an accessory pathway for disposal of microbial phenylpropionic acid as phenylpropionylglycine (Fig. 4c). Supporting our data, previous studies have shown accumulation of phenylpropionylglycine in urine of humans with MCAD deficiency[26]. Although the nature of residual hippuric acid production in MCAD$^{-/-}$ mice is unclear, we speculate that it may involve peroxisomal acyl-CoA oxidases[27].

From the data presented here, we propose that phenylalanine is converted to hippuric acid via a pathway involving bacterial transformations in the gut and host metabolism (Supplementary Fig. 3). In the absence of a functional MCAD gene, phenylpropionic acid is shunted through an accessory pathway with glycine *N*-acyltransferase (GLYAT), yielding phenylpropionylglycine (Fig. 4c). Consistent with this, GF MCAD$^{-/-}$ mice gavaged with $d_9$-phenylpropionic acid (Fig. 4d) accumulate labeled phenylpropionylglycine, whereas GF mice with functional MCAD do not (Fig. 4e), routing this molecule instead to hippuric acid (Fig. 2c).

### MCAD oxidizes additional microbial metabolites
Having characterized the role of MCAD in phenylpropionic acid metabolism to hippuric acid, we next asked whether additional microbial metabolites are metabolized by MCAD. To address this,

we performed untargeted metabolomics on urine from GF and *C. sporogenes*-colonized MCAD$^{+/+}$ and MCAD$^{-/-}$ mice (Supplementary Data 3). To focus on microbial metabolites, we limited our investigation to compounds that were dependent on *C. sporogenes* colonization (elevated post-colonization compared to germ-free). From these analyses, we identified 29 LC-MS features that differed based on host genotype, most elevated in the urine of MCAD$^{-/-}$ mice (Fig. 5a, Supplementary Fig. 4a). By searching our in-house metabolite database and publicly available MS/MS databases, we were able to derive presumptive structural identifications of four compounds which we subsequently verified by comparison with authentic standards (Fig. 5b, Supplementary Data 4). We were reassured to find phenylpropionylglycine elevated in the urine of MCAD$^{-/-}$ mice, consistent with the role of MCAD in phenylpropionic acid to hippuric acid metabolism shown in Fig. 4. Cinnamoylglycine, the glycine conjugate of cinnamic acid, was decreased in the urine of MCAD$^{-/-}$ mice (Supplementary Fig. 4a) and is likely an intermediate in MCAD-dependent metabolism of phenylpropionic acid (Supplementary Fig. 3). The identification of two additional metabolites, 3-(4-hydroxyphenyl)propionic acid (4-OH-PPA) and isocaproylglycine was unexpected. 4-OH-PPA is the reductive product of tyrosine metabolism by *C. sporogenes*, generated by the same pathway for phenylpropionic acid metabolism (Supplementary Fig. 1), whereas isocaproylglycine bears resemblance to the *C. sporogenes* end product of leucine metabolism, isocaproic acid[5].

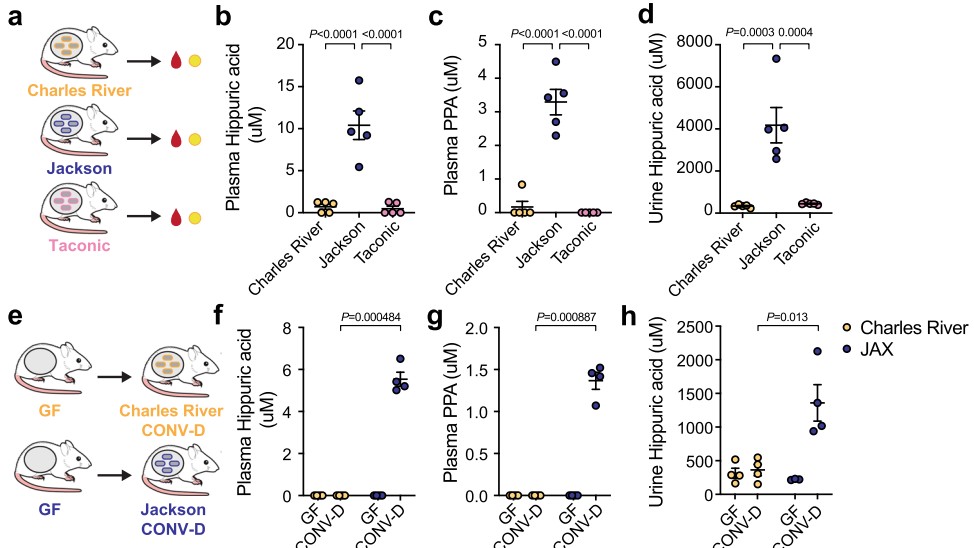

**Fig. 3 | Circulating hippuric acid and phenylpropionic acid in conventional mice differ by vendor. a** Plasma and urine samples were obtained from Swiss Webster conventional mice from Charles River, Jackson, or Taconic Laboratories. Hippuric acid (**b**) and phenylpropionic acid (**c**) levels were significantly higher in mice from Jackson labs compared to Charles River or Taconic. **d** Hippuric acid was elevated in urine of conventional mice from Jackson Laboratories (normalized to concentration of creatinine). For **b**–**d**: One-way ANOVA with Tukey's post-hoc comparisons, $n = 5$ mice/group, mean ± s.e.m. shown Plasma hippuric acid $F_{(2,12)} = 31.48$, plasma PPA $F_{(2,12)} = 60.09$, urine hippuric acid $F_{(2,12)} = 20.30$. **e** Cecal contents derived from conventional Swiss Webster mice obtained from Charles River or Jackson Laboratories were transplanted to germ-free recipients; plasma and urine samples were obtained. GF, germ-free; CONV-D, conventionalized. After cecal transplantation, plasma hippuric acid (**f**) and phenylpropionic acid (**g**) were significantly higher in Jackson labs recipients compared to Charles River (unpaired two-tailed $t$-tests). **h** Hippuric acid was significantly higher in urine of mice with cecal contents from Jackson labs mice (two-sided Mann-Whitney test). For **f**–**h**: $n = 4$ mice/group, mean ± s.e.m. shown. The mouse schematic in **a**, **e** was adapted from ref. [17].

In our initial GC-TOF metabolomics, we identified 4-hydroxyhippuric acid (4-OH-hippuric acid) in urine as dependent on *fldC* (Supplementary Fig. 4b) and, together with the untargeted MCAD$^{-/-}$ metabolomics analysis, hypothesized that 4-OH-PPA undergoes a similar MCAD-dependent transformation as phenylpropionic acid (Fig. 5c). We used stable isotope dilution LC-MS to quantify 4-OH-PPA, 4-OH-hippuric acid, and 3-(4-hydroxyphenyl)propionylglycine (4-OH-PPG, chemically synthesized for this study) in the urine of germ-free and *C. sporogenes*-colonized MCAD$^{-/-}$ and MCAD$^{+/+}$ mice. Consistent with the role of MCAD in this pathway, urine 4-OH-hippuric acid is elevated in MCAD$^{+/+}$ mice whereas 4-OH-PPA is elevated in MCAD$^{-/-}$ mice, and 4-OH-PPG trends towards a significant increase in MCAD$^{-/-}$ mice (Fig. 5d). These results support a model in which 4-OH-PPA is metabolized via MCAD in a similar manner as phenylpropionic acid. The MCAD pathway does not proceed to completion, as both 4-OH-PPA (the precursor) and 4-OH-PPG (the alternative pathway product) are detected in urine of colonized MCAD$^{+/+}$ mice (Fig. 5d). These results identify microbially-produced 4-OH-PPA as an additional substrate for host β-oxidation via MCAD.

We predicted that microbial isocaproic acid may also be metabolized by host MCAD (Fig. 5e). Since the chain-shortened and glycine-conjugated MCAD product of isocaproic acid, isobutyrylglycine (Fig. 5e), co-eluted with butyrylglycine, we quantified the product that we hypothesized would accumulate in the absence of MCAD, isocaproylglycine. Consistent with our expectations, isocaproylglycine levels were elevated in MCAD$^{-/-}$ compared to MCAD$^{+/+}$ mice by LC-MS (Fig. 5f). In contrast to phenylpropionylglyine (Fig. 4b) and 4-OH-PPG (Fig. 5d), germ-free MCAD$^{-/-}$ mice have appreciable levels of isocaproylglycine, suggesting that endogenous host pathways produce this metabolite. However, isocaproylglycine levels are significantly higher in MCAD$^{-/-}$ mice upon colonization ($P < 0.0001$), indicating that isocaproic acid supplied by *C. sporogenes* in the gut elevates levels of isocaproylglycine in MCAD-deficient mice (Fig. 5f). These data suggest isocaproic acid as a third product of bacterial metabolism metabolized by host MCAD.

## Discussion

The gut microbiome, through its metabolism of diet- and host-derived compounds, expands the biochemical inventory of the host. Here, we focused on a specific microbial pathway, aromatic amino acid metabolism encoded by the *fld* locus, and identified a host-microbe co-metabolic pathway that contributes to one of the most abundant urine metabolites in mammals, hippuric acid. Our findings indicate that phenylalanine is converted to phenylpropionate in the gut, which is then absorbed by the host and subjected to β-oxidation via MCAD before being conjugated to glycine and eliminated in the urine. We also find that host MCAD participates in metabolism of other microbially derived substrates such as 3-(4-hydroxyphenyl)propionic acid (also known as desaminotyrosine) and isocaproic acid, a reductive pathway product of leucine metabolism. Together, our data reveal host β-oxidation through MCAD as a previously unidentified mechanism for host metabolism of gut microbiota-derived molecules (Fig. 5g).

Several host enzymes acting on microbiota-dependent host metabolites have been well-characterized. These include cytochrome P450s, flavin monooxygenases, alcohol dehydrogenases and monoamine oxidases which introduce reactive, or polar groups into compounds (phase I metabolism), whereafter activated metabolites are conjugated with glutathione, glycine, sulfate, and glucuronic acid species (phase II). While glycine conjugation is typically considered a detoxification mechanism by increasing water solubility to facilitate urinary excretion, more recently it has been suggested that glycine conjugation may regulate systemic levels of aromatic amino acids, such as those utilized as neurotransmitters[28]. However, apart from butyrate metabolism, host β-oxidation has not been considered an important mechanism for the metabolism of microbial metabolites. Our identification of phenylpropionic acid, 3-(4-hydroxyphenyl)propionic acid, and isocaproic acid as likely substrates for host MCAD are in agreement with its known preference for fatty acids ranging in length from $C_6$-$C_{12}$[29]. Further corroborating our findings are the demonstration of activity for purified MCAD with phenylpropionyl-CoA as substrate[26] and studies in MCAD-deficient patients, which

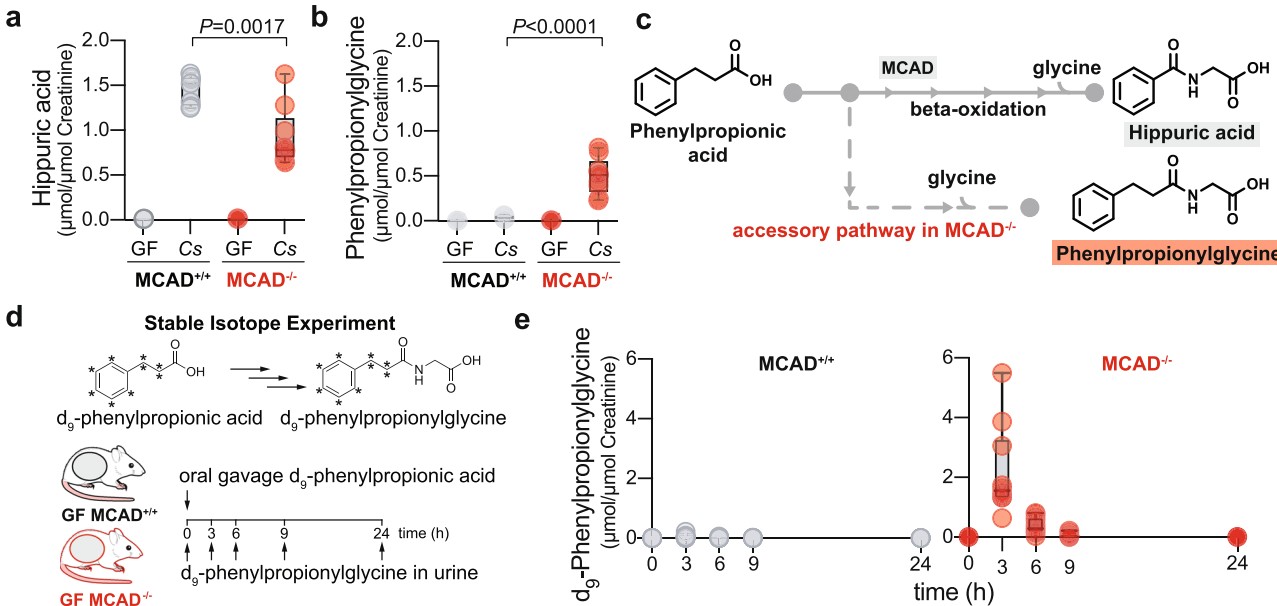

**Fig. 4 | Host medium chain acyl-CoA dehydrogenase is involved in phenyl-propionic acid metabolism to hippuric acid. a, b** Urine was collected from MCAD[+/+] or MCAD[−/−] mice either in the germ-free (GF) state or following colonization with wild-type *C. sporogenes* (*Cs*). Hippuric acid (**a**) and phenylpropionylglycine (**b**) levels were determined in the urine by stable isotope dilution mass spectrometry and normalized to urinary creatinine levels (*n* = 4 mice/GF group, *n* = 7 MCAD[+/+] *Cs*-colonized, *n* = 9 MCAD[−/−] *Cs*-colonized; unpaired two-sided Student's *t*-tests). **c** Model for host metabolism of phenylpropionic acid involving MCAD, and

presence of an accessory pathway converting phenylpropionic acid to phenyl-propionylglycine. **d** Germ-free mice were orally administered isotopically labeled phenylpropionic acid and labeled phenylpropionylglycine was quantified in urine over time. **e** Urinary levels of d9-phenylpropionylglycine after d9-phenylpropionic acid gavage in GF MCAD[−/−] or MCAD[+/+] mice (*n* = 12 mice/group) by stable isotope dilution mass spectrometry and normalized to urinary creatinine levels. For **a, b, e**: boxes denote the median with inter-quartile distance, whiskers maxima and minima. The mouse schematic in **d** was adapted from ref. [17].

identified increased urinary levels of phenylpropionylglycine, decreased hippuric acid[26, 30] and increased isocaproylglycine[31]. Although it has been suggested that metabolic activity in the gut is responsible for the production of these metabolites[32, 33], our results with gnotobiotic mice and bacterial genetics now provide definitive evidence. Our results also implicate MCAD in the growing list of host enzymes that generate high-abundance circulating metabolites via co-metabolism with gut bacteria.

An interesting theme that emerges from our analysis of host-microbe co-metabolism is that several microbial metabolites are excreted as their glycine conjugates. This is presumably due to activity of host glycine N-acyltransferase (GLYAT), a mitochondrial enzyme that uses acyl-CoAs and glycine as co-substrates, forming acylglycines and CoASH. The metabolic rationale for glycine conjugation in mammals is not entirely clear, but several hypotheses have been suggested (summarized in ref. [28]): (1) Glycine conjugation changes the lipophili-city of aromatic metabolites such as benzoic acid, limiting their toxicity. (2) Glycine conjugation via GLYAT recycles CoASH which is required for a number of critically important biochemical activities. (3) Reduced host glycine levels resulting from conjugation with excess aromatic metabolites might represent a molecular circuit influencing diet preference.

Hippuric acid in human urine has been postulated to be microbially-derived[4,16,20,34–36], but neither a microbe nor microbial gene responsible for its production has been defined. Our results in gno-tobiotic mice colonized by wild-type or *fldC* mutant *C. sporogenes* suggest that reductive metabolism of phenylalanine is a major source of hippuric acid in the host. However, previous studies have suggested that hippuric acid arises from microbiota-independent mechanisms (e.g., ingestion of benzoic acid or toluene exposure) or through microbial degradation of dietary polyphenols and flavonoids such as those present in coffee and tea[19]. Indeed extensive literature has linked hippuric acid excretion in humans to consumption of teas, juices, and fruits known to be rich in polyphenols (reviewed in ref. [19]). Thus, it

seems likely that the extent of hippuric acid excretion in individuals is multifactorial, representing non-microbial dietary contributions and those from microbial metabolism of phenylalanine (as we have shown) and of dietary polyphenols. Stable isotope tracing with labeled food components will likely be required to tease apart the relative con-tributions of these dietary and microbial sources to hippuric acid production. Along these lines, a recent study feeding isotope labeled food components to mice demonstrated that dietary protein is an important precursor for systemic hippuric acid with an estimated contribution of 33%[37].

Beyond identifying MCAD as a host enzyme responsible for metabolizing microbial metabolites, our work has several other important implications: first, while hippuric acid is ubiquitous and has not been linked to adverse health outcomes in humans, phenylace-tylglycine (phenylacetylglutamine in humans), the glycine-conjugated product of oxidative phenylalanine metabolism, is associated with an increased risk for cardiac disease[8]. Therefore, the reductive conversion of phenylalanine via phenylpropionic acid to hippuric acid, which is largely regarded as non-toxic in humans, represents a comparably healthy alternative to the oxidative pathway for phenylacetate pro-duction. Second, a microbial pathway that functions at high flux, con-verting phenylalanine to a non-toxic end-product could be of considerable interest with respect to patients with phenylketonuria where their blood phenylalanine levels are elevated. Along these lines, previous groups have shown that *Escherichia coli* engineered to degrade phenylalanine in the gut can reduce levels in the blood. Identifying naturally occurring gut bacteria capable of degrading phenylalanine in the gut represents an alternative to engineered strains.

This work presents a ubiquitous mammalian metabolic pathway as a novel avenue by which the host metabolizes several compounds generated by the gut microbiota. In addition to phenylpropionic acid, 3-(4-hydroxyphenyl)propionic acid, and isocaproic acid, several uni-dentified glycine-conjugated products of bacterial metabolism accu-mulate in the absence of functional MCAD, indicating that under

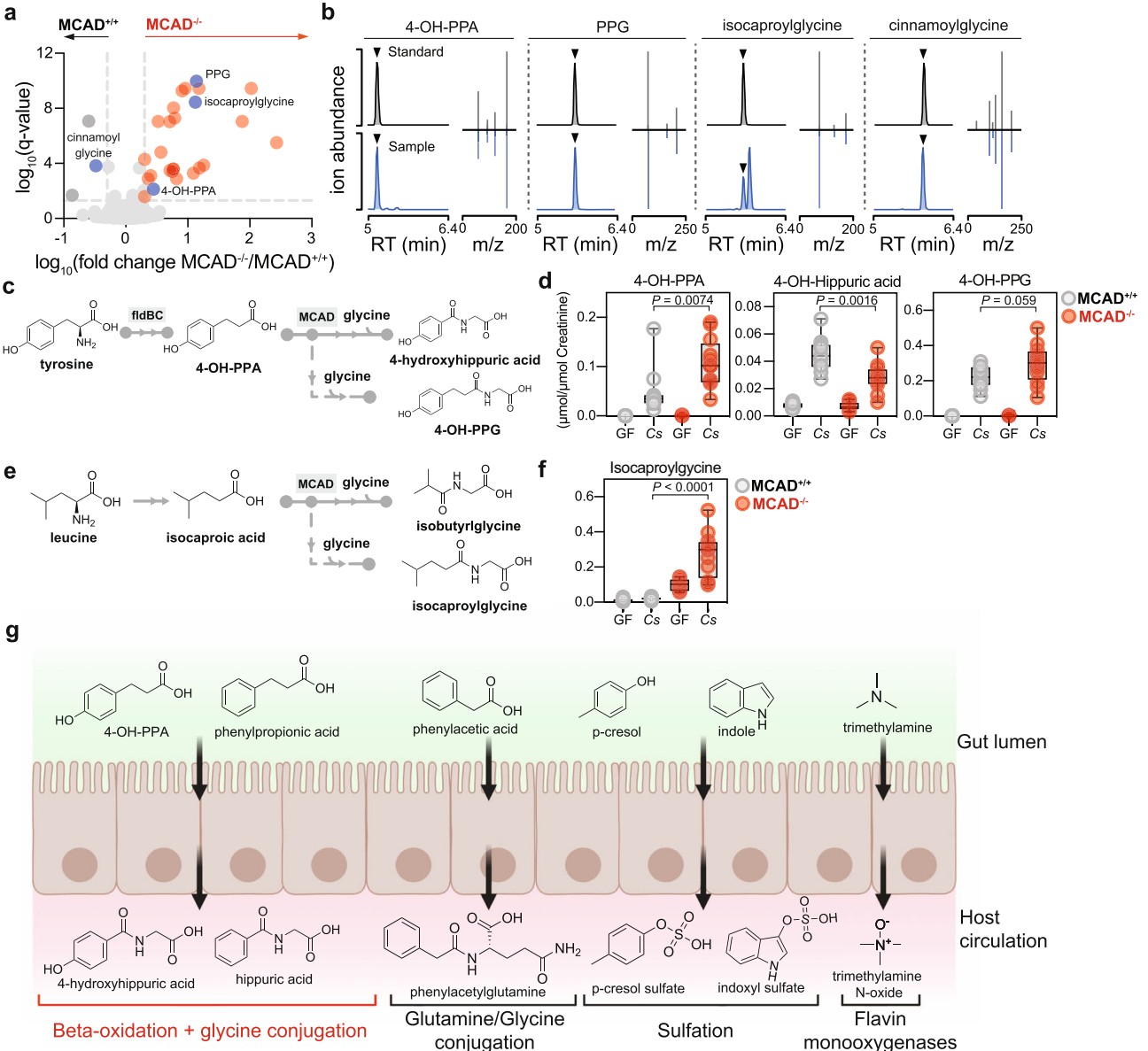

**Fig. 5 | Host MCAD metabolizes additional products of microbial metabolism.** **a** Urine metabolites dependent on *C. sporogenes* colonization that are significantly different between MCAD⁻ᐟ⁻ (positive fold change, red) and MCAD⁺ᐟ⁺ mice (negative, gray). Metabolites are colored if significant (*q* value < 0.05) and exhibit a greater than two-fold change (paired *t*-tests with two-stage linear step-up procedure of Benjamini, Krieger, and Yekutieli). Named metabolites are colored in blue. **b** Total ion chromatograms and MS/MS fragmentation spectra for validation of four predicted metabolites in **a**. Standards are shown above; biological samples, below in blue. **c** Conversion of tyrosine to 3-(4-hydroxyphenyl)propionic acid by *C. sporogenes* and predicted conversion to 4-hydroxyhippuric acid via host MCAD and glycine conjugation yielding 3-(4-hydroxyphenyl)propionylglycine in MCAD⁻ᐟ⁻. **d** 4-OH-PPA and 4-OH-PPG accumulate in the absence of MCAD, whereas 4-OH-hippuric acid is higher in MCAD⁺ᐟ⁺ mice (4-OH-PPA and 4-OH-hippuric acid:

unpaired two-tailed Student's *t*-test, 4-OH-PPG: two-sided Mann-Whitney). **e** *C. sporogenes* metabolism of leucine produces isocaproic acid; predicted products based on MCAD activity in the host. **f** Isocaproylglycine accumulates in MCAD-deficient mice, indicating isocaproic acid undergoes host β-oxidation under homeostatic conditions (unpaired two-tailed Student's *t*-tests). For **d** and **f**: *n* = 14 mice/GF group, *n* = 11 MCAD⁺ᐟ⁺ *Cs*-colonized, *n* = 12 MCAD⁻ᐟ⁻ *Cs*-colonized mice; data were combined across two independent experiments. Boxes denote the median with inter-quartile distance, whiskers indicate maxima and minima. **g** Four previously demonstrated compounds in host circulation are the product of canonical host phase I (flavin monooxygenases) and phase II (sulfation, glutamine or glycine conjugation) metabolism of microbially-produced metabolites. Host β-oxidation (red) by MCAD generates hippuric acid and 4-hydroxyhippuric acid, two high-abundance compounds in the host.

homeostatic conditions, these molecules undergo β-oxidation prior to glycine conjugation. Host MCAD thus represents a broad and previously unsuspected mechanism for metabolism of microbial metabolites.

## Methods

### Animal husbandry

All animal experiments were conducted under a protocol approved by the Stanford University Institutional Animal Care and Use Committee.

All mouse experiments were conducted in a dedicated gnotobiotic facility using flexible film aseptic breeder isolators or experimental isolators, both from Class Biologically Clean (Madison, WI). In some shorter experiments (<1 week), an IsoCage P rack system with cage-level HEPA filtration (Tecniplast, Buguggiate, Italy) was used. Animals were maintained on a 12-h light/dark cycle and fed standard chow (LabDiet 5K67) and water *ad libitum*. Sterility of animals was verified by bi-weekly anaerobic and aerobic cell culture of feces for representative animals from each isolator and PCR of the 16S rRNA locus. Initial

untargeted metabolomics mouse experiments were performed on gnotobiotic Swiss Webster germ-free mice (male, 8–12 weeks of age, $n = 5$ per group) originally obtained from Taconic Biosciences. Additional experiments were performed on MCAD$^{-/-}$ mice (B6;129P2-Acadm$^{tm1Uab}$/Mmmh)[25] which were originally obtained as embryos from the Mutant Mouse Resource & Research Center (MMRRC, Cat. # 011588-MU) and transferred to the National Gnotobiotic Rodent Resources Center (NGRRC) where they were re-derived by embryo transfer into germ-free C57/BL6 mice. Heterozygous germ-free MCAD$^{+/-}$ mice were transferred to our gnotobiotic mouse facility at Stanford where the colony was expanded, and breeding was maintained as homozygous MCAD$^{-/-}$ or MCAD$^{+/+}$ isogenic background breeding pairs, the latter serving as controls. Conventional Swiss Webster mice were obtained from Taconic, Jackson Laboratories, and Charles River Laboratories. Mice were maintained on standard chow and housed in flexible film isolators. Each experiment comparing colonization state and/or host genotype was performed with sex-matched animals. Both male and female mice were used; the sex of animals used in a given experiment was determined by animal availability within our gnotobiotic facility.

## Bacterial culture

*Clostridium sporogenes* ATCC 15579 was routinely cultured in Reinforced Clostridial Media (RCM, 10 g/L peptone, 10 g/L beef extract, 3 g/L yeast extract, 5 g/L dextrose, 5 g/L sodium chloride, 1 g/L soluble starch, 0.5 g/L cysteine hydrochloride, 3 g/L sodium acetate, 0.5 g/L agar) at 37 °C in an anaerobic chamber from Coy Laboratories under an atmosphere of 5% hydrogen, 10% $CO_2$ and 85% $N_2$. The *C. sporogenes fldC* mutant, which harbors a ClosTron insertional mutation, was generated previously[17] and was cultured in identical conditions to the wild-type strain. Bacterial cells were stored at −80 ˚C as anaerobically prepared 25% v/v glycerol stocks sealed in 12 × 32 mm glass crimp top vials. Liquid cultures were transferred to sterile 2 mL cryovials with inner threading prior to gavage. All manipulations with *C. sporogenes* were performed in the anaerobic chamber and with reagents and media pre-reduced for at least 48 h.

## MCAD$^{-/-}$ genotyping

Mouse MCAD genotype was confirmed with DNA PCR targeting the neomycin insert, which distinguishes between MCAD$^{+/+}$ and MCAD$^{+/-}$ or MCAD$^{-/-}$ as well as RT-PCR to detect the MCAD transcript. Qiagen DNeasy Blood and Tissue kit (cat #69506) was used to extract DNA from ear punch. Primers used were: NeoF 5′- CATTCGACCACCAAGC-GAAACATC-3′, NeoR 5′-ATATCACGGGTAGCCAACGCTATG-3′[25]. The product was analyzed on a 3% agarose gel with an expected band size of 289 bp for heterozygous and knockout genotype, and no product for wild-type genotype. RNA was extracted from tail clip with Qiagen RNeasy Mini kit (cat #74104) and RT-PCR performed with Qiagen One Step RT-PCR kit (cat #210212). Primers used were: M11588F 5′- AT GTGGCGGCCATTAAGACCAAAG-3′, M11588R 5′- GCTGATTGGCAATG TCTCCAGCAA-3′. Products were imaged on a 3% agarose gel; both MCAD$^{+/+}$ and MCAD$^{+/-}$ animals have a single 550 bp product, while MCAD$^{-/-}$ animals have a ladder effect, with approximate product sizes of 700, 800, 1000 bp and larger.

## Primary metabolite profiling by GC-TOF

Germ-free Swiss Webster mice (male, 8–12 weeks age, $n = 5$ per group) were colonized with either wild-type *Clostridium sporogenes* ATCC 15579 or its *fldC* mutant strain[17] ($1 \times 10^7$ CFU per mouse, 200 μL). Four weeks following colonization, urine and feces were collected, then mice were euthanized and blood was collected by cardiac puncture and cecal contents were harvested. Serum was obtained from whole blood using BD microtainers (cat #365967) following the manufacturers guidelines, then snap frozen in liquid nitrogen and stored at −80 °C. Urine, feces, and cecal contents were transferred to 2 mL

screw-cap tubes, and snap frozen in liquid nitrogen immediately after collection, then stored at −80 °C. Samples were shipped on dry ice to the West Coast Metabolomics Center at University of California, Davis where metabolites were extracted, derivatized, and analyzed by gas chromatography time of flight mass spectrometry (GC-TOF-MS).

Samples (serum 30 μL, feces and cecal contents -20 mg, and urine 20 μL) were thawed at 4 °C, then metabolites were extracted, derivatized, and analyzed by GC-MS using standard protocols, as described previously[38]. Reported data (Supplementary Data 1) represent peak heights for each corresponding quantification ion normalized to the sum of the peak heights for all named metabolites within a treatment group.

## GC-TOF metabolomics data analysis

Metabolites that were identified in the shotgun metabolomics platform were included in downstream analysis. For determination of statistical significance, discovery was determined using the two-stage linear step-up procedure of Benjamini, Krieger and Yekutieli, with $Q = 5\%$, implemented in GraphPad Prism v.8.4.1. Peak areas were log$_{10}$-normalized and each metabolite was analyzed individually without assuming a consistent SD with a total of 225 $t$-tests for each host compartment.

## Stable isotope tracing in gnotobiotic mice

For phenylpropionic acid stable isotope tracing experiments, germ-free mice were administered 200 μL of a sterile-filtered 50 mM solution of d$_9$-phenylpropionic acid (C/D/N isotopes Cat. # D-5666) dissolved in water by oral gavage using a 1 mL tuberculin syringe equipped with a 18–19 gauge gavage needle (Popper 7900). Urine was collected at $t = 0$, 3, 6, 9, 24 h following gavage and d$_5$-hippuric acid was quantified in the urine by LC-MS (see LC-MS method section for details).

For phenylalanine stable isotope tracing experiments, mice were administered 200 μL of a sterile-filtered 50 mM solution of d$_5$-phenylalanine (Sigma Cat. # 615870) dissolved in water by oral gavage using a 1 mL tuberculin syringe equipped with a 18–19 gauge gavage needle (Popper 7900). Urine was collected at $t = 0$, 3, 6, 9, 24 h following gavage and d$_5$-hippuric acid was quantified in the urine by LC-MS (see LC-MS method section for details). Stable isotope tracing was first performed on germ-free mice to test endogenous conversion of labeled phenylalanine. Subsequently, the same mice were colonized with either wild-type or *fldC* mutant *Clostridium sporogenes* by oral gavage (200 μL, -1 × 10$^7$ CFU) and stable isotope tracing was performed one week following colonization.

## Stable isotope tracing in MCAD$^{-/-}$ mice

MCAD$^{-/-}$ mice were re-derived as germ-free for the purpose of this study. MCAD$^{-/-}$ mice or MCAD$^{+/+}$ controls were maintained in sterile gnotobiotic isolators or cages with HEPA-filtered air supply. Stable isotope tracing was performed as described above: urine from GF MCAD$^{-/-}$ and MCAD$^{+/+}$ mice was collected under sterile conditions immediately prior to gavage with 50 mM d$_9$-phenylpropionic acid and 3, 6, 9, and 24 h thereafter. Seven days subsequent, mice were mono-colonized with wild-type *C. sporogenes* by gavage of 200 μL saturated overnight culture. One week after colonization, d$_9$-phenylpropionic acid gavage and urine collection at 0, 3, 6, 9, and 24 h were repeated. *C. sporogenes* colonization was verified with serial dilution plating. Urinary creatinine, hippuric acid, labeled and unlabeled phenylpropionylglycine were quantified by LC-MS (see LC-MS method section for details).

## Liquid chromatography mass spectrometry (LC-MS)

During the course of this study, our laboratories shifted from primarily using a triple quadrupole mass spectrometer (Agilent 6470) to a quadrupole time-of-flight (Q-TOF) mass spectrometer (Agilent 6545XT), therefore methods for LC-MS quantitation of metabolites

differed slightly, as described below. LC-MS grade solvents were purchased from Fisher Scientific (Pittsburgh, PA); $d_3$-creatinine, from Cambridge Isotope Laboratories (Tewksbury, MA). No attempt was made to correct for natural abundance of heavy isotopes, as the deuterated compounds we chose for administration to mice and as internal standards were sufficiently distinct such that analytes of interest would be negligibly affected (e.g. isotopic purity of $d_2$-hippuric acid contains <0.1% hippuric acid and $d_5$- hippuric acid).

## Quantification of hippuric acid in GF, WT, and fldC-colonized mice

For confirmation of hippuric acid levels in the urine of germ-free, WT, and *fldC*-colonized mice (Fig. 2a), and for stable isotope tracing of $d_9$-phenylpropionic acid and $d_5$-phenylalanine in Swiss Webster mice (Fig. 2c, e), we used stable isotope dilution mass spectrometry with an Agilent 6470 triple quadrupole mass spectrometer. Unlabeled hippuric acid was purchased from Fisher Scientific (Pittsburgh, PA). For sample preparation, 5 μL of urine was diluted with 45 μL of LC-MS grade water, then 50 μL of 6% v/v sulfosalicylic acid in water was added to precipitate proteins. After centrifugation at $12,000 \times g$ for 5 min, 50 μL of supernatant was transferred to a new tube and 75 μL of 50% acetonitrile containing 2.5 μM $d_5$-hippuric acid (Cambridge Isotope Laboratories, Tewksbury, MA) as an internal standard was added. For experiments where $d_5$-hippuric acid was to be detected in mouse urine (e.g., for stable isotope tracing), 2,2-$d_2$-hippuric acid (C/D/N isotopes, Pointe-Claire, Quebec, Canada) was used as an internal standard, following its corresponding SRM transition (180.1→136.1). Tubes were vortexed briefly to mix, then centrifuged and transferred to a 96-well autosampler plate fitted with a silicone cap map and stored at 4 °C prior to analysis. Compounds were separated using an Agilent 1290 Infinity II UPLC equipped with an Agilent RRHD 1.8 μm particle size C18 column (2.1 × 50 mm) and detected with an Agilent 6470 Triple Quad mass spectrometer equipped with a jet stream electrospray ionization source. Eluent A consisted of 0.1% formic acid (v/v) in water and eluent B consisted of 0.1% formic acid (v/v) in acetonitrile. Samples (5 μL) were injected via refrigerated autosampler into mobile phase and chromatographic separation at 40 °C was achieved at a flow rate of 0.5 mL min$^{-1}$ using the following gradient: 0–1.5 min, 10–12% B; 1.5–1.75 min, 12–90% B; 1.75–2.75 min, 90% B; 2.75–3 min, 90–10% B; 3–4.75 min, 10% B. The first 0.5 min was diverted to waste, then hippuric acid isotopes were detected in negative ionization mode using the specific SRM transitions for hippuric acid (178.1→134.0) and internal standard, $d_5$-hippuric acid (183.1→139.1) with dwell time of 200 ms, fragmentor voltage of 100 V, and collision energy of 9 V. Peak areas were normalized using the internal standard, and concentrations were determined by comparison to calibration curves prepared from dilution series of the authentic standard (hippuric acid, Sigma Aldrich, St. Louis, MO). Electrospray ionization parameters included gas temperature of 300 °C, gas flow of 6 L/min, sheath gas temperature of 350 °C, sheath gas flow of 12 L/min, and capillary voltage of 2500 V.

## Quantification of hippuric acid and phenylpropionylglycine in MCAD$^{+/+}$ and MCAD$^{-/-}$ mice

For quantitation of hippuric acid and *N*-(3-phenylpropionyl)glycine (PPG) in MCAD$^{+/+}$ and MCAD$^{-/-}$ gnotobiotic mouse urine (Fig. 3) we used stable isotope dilution with an Agilent 6545XT Q-TOF. We also quantified urine creatinine such that we could normalize urine solutes across animals. Mouse urine samples (5 μL) were mixed with internal standards (10 μL, $d_3$-creatinine and 2,2-$d_2$-hippuric acid, 0.5 mM each) and diluted with LC-MS grade water (15 μL) in 96-well V-bottom plates from USA scientific. 90 μL of acetonitrile/methanol 3:1 mixture was added to precipitate proteins. The solution was mixed five times by pipetting, then the plate was centrifuged at $5000 \times g$ for 10 min at 4 °C. Supernatant (40 μL) was transferred to 96-well autosampler plates containing 160 μL water, mixed well and a cap mat was placed on the

plates. Samples were analyzed by LC-MS using an Agilent 1290 Infinity II UPLC equipped with a Waters 1.7-μm particle size BEH C18 column (2.1 × 100 mm) and detected using an Agilent 6545XT Q-TOF equipped with a dual jet stream electrospray ionization source operating under extended dynamic range (EDR 1700 *m/z*). Eluent A consisted of 0.1% acetic acid (v/v) in water and eluent B consisted of 0.1% acetic acid (v/v) in acetonitrile. Samples (0.5 μL) were injected via refrigerated autosampler into mobile phase and chromatographic separation at 40 °C was achieved at a flow rate of 0.4 mL min$^{-1}$ using the following gradient: 0–2 min, 1% B; 2–6 min, 1–98% B; 6–7 min, 98% B; 7–7.5 min, 98–1% B; 7.5–11 min, 1% B. Quantitation of creatinine was conducted in ESI positive mode using $d_3$-creatinine as an internal standard. Quantitation of $d_5$-hippuric acid and $d_5$-*N*-(3-phenylpropionyl)glycine were conducted in ESI negative mode using 2,2-$d_2$-hippuric acid as internal standard. Electrospray ionization parameters included fragmentor voltage of 140 V, gas temperature of 300 °C, sheath gas temperature of 275 °C, and capillary voltage of 4000 V. Peak areas were normalized using the internal standard ($d_2$-hippuric acid for hippuric acid and phenylpropionylglycine, $d_3$-creatinine for creatinine), and concentrations were determined by comparison to calibration curves prepared from dilution series of authentic standards (creatinine, hippuric acid, and phenylpropionylglycine, Toronto Research Chemicals, North York, Ontario, Canada) spiked in 1× PBS solution as a surrogate matrix. The calibration range for creatinine was 0.029–30 mM, and 0.020–20 mM for hippuric acid and phenylpropionylglycine. A linear calibration regression model was applied to creatinine and hippuric acid; a quadratic regression model was applied to phenylpropionylglycine, using a 1/x weighted least-squares regression algorithm. $R^2 = 0.996$ for creatinine, and 1.000 for hippuric acid and phenylpropionylglycine.

For $d_9$-phenylpropionic acid stable isotope tracing experiments, $d_9$-*N*-(3-phenylpropionyl)glycine was quantitated. Urine samples were prepared and analyzed according to the above protocol except that the supernatant (20 μL) was diluted in 96-well autosampler plates with 180 μL water. Quantification of urinary organic acids and related metabolites. For quantitation of adipic acid, suberic acid, and hexanoylglycine, mouse urine samples (2.5 μL) were mixed with internal standards (15 μL, $d_3$-creatinine and $d_2$-isovaleric acid, 0.2 mM each) and diluted with LC-MS grade water (7.5 μL) in 96-well V-bottom plates from USA scientific. Acetonitrile/methanol 3:1 mixture (75 μL) was added to precipitate proteins. The solution was mixed five times by pipetting, then the plate was centrifuged at $15,000 \times g$ for 5 min at room temperature. Supernatant (10 μL) was transferred to 96-well autosampler plates containing 90 μL water, mixed well and a cap mat was placed on the plates. Samples were analyzed by LC-MS using an Agilent 1290 Infinity II UPLC equipped with a Waters 1.7-μm particle size BEH C18 column (2.1 × 100 mm) and detected using an Agilent 6545XT Q-TOF equipped with a dual jet stream electrospray ionization source operating under extended dynamic range (EDR 1700 *m/z*). Eluent A consisted of 0.1% acetic acid (v/v) in water and eluent B consisted of 0.1% acetic acid (v/v) in methanol. Samples (2 μL) were injected via refrigerated autosampler into mobile phase and chromatographic separation at 60 °C was achieved at a flow rate of 0.4 mL min$^{-1}$ using the following gradient: 0–1 min, 1% B; 1–15.5 min, 1–45% B; 15.5–15.6 min, 45–99% B; 15.6–16.9 min, 99% B, 16.9–17 min, 99–1% B; 17–19 min, 1% B. Quantitation of creatinine was conducted in ESI positive mode using $d_3$-creatinine as an internal standard. Quantitation of adipic acid, suberic acid, and hexanoylglycine, was conducted in ESI negative mode using $d_2$-isovaleric acid as internal standard. Electrospray ionization parameters included fragmentor voltage of 70 V, gas temperature of 250 °C, sheath gas temperature of 250 °C, and capillary voltage of 4000 V. Peak areas were normalized using the internal standard and concentrations were determined by comparison to calibration curves prepared from dilution series of authentic standards. The calibration range is 0.005–40 mM for creatinine,

0.010–10 mM for adipic acid, 0.020–10 mM for suberic acid, and 0.005–20 mM for hexanoylglycine. Linear calibration regression model was applied to adipic acid and suberic acid, and quadratic regression model was applied to creatinine and hexanoylglycine, using $1/x$ weighed least-squares regression algorithm. $R^2 = 0.996$, 0.998, 0.999, and 1.000 for adipic acid, suberic acid, hexanoylglycine, and creatinine.

### Cecal transplants into germ-free mice

Cecal contents were harvested from Swiss Webster mice from different vendors (Taconic, Jackson Laboratories, Charles River Laboratories), snap frozen in liquid nitrogen, and stored at −80 °C. Cecal contents from $n = 5$ mice were thawed on ice, pooled, and orally gavaged into recipient germ-free Swiss Webster mice. Mice were then maintained on standard chow in an Isocage P cage rack system for 2 weeks until urine and plasma samples were collected for LC-MS analysis.

### Quantification of hippuric acid and phenylpropionic acid in plasma and urine of conventional Swiss Webster and cecal content transplanted germ free mice

Metabolites were extracted from plasma and urine samples, derivatized with 3-nitrophenylhydrazine, and analyzed by LC-MS as described previously[39].

### Untargeted metabolomics analysis of MCAD[+/+] vs. MCAD[−/−] mice

Untargeted metabolomics was performed on mouse urine samples collected before administration of $d_5$-phenylalanine or $d_9$-phenylpropionic acid ($t = 0$ h). Mouse urine samples (2.5 µL) were diluted with LC-MS grade water (5 µL) and mixed with internal standard (7.5 µL, $d_3$-creatinine, 1 mM; $d_9$-phenylpropionic acid, 0.2 mM; 2,2-$d_2$-hippuric acid, 0.2 mM), then 60 µL of extraction solvent containing reference standards ($d_7$-glucose, $d_3$-methionine, L-4-hydroxyphenyl-$d_4$-alanine, $d_5$-hippuric acid, $d_5$-tryptophan, $d_3$-leucine, di-n-octyl phthalate-3,4,5,6-d4, $d_{19}$-decanoic acid, $d_{15}$-octanoic acid, $d_{27}$-tetradecanoic acid, 2-flurophenylglycine, $d_9$-carnitine in methanol) was added to precipitate proteins. The solution was vortexed, then the plate was centrifuged at $15,000 \times g$ for 5 min at 4 °C. Supernatant (60 µL) was transferred to a new plate and 20 µL of each was diluted with 60 µL LC-MS grade water. The rest of the supernatant was stored at −80 °C as a backup. Three procedure blanks were included using the same preparation method with LC-MS water (2.5 µL) instead of urine sample. Quality controls (QC) were pooled from each LC-MS-ready sample.

Samples (0.5 µL) were subjected to LC-MS/MS analysis under both ESI positive mode and negative mode. Negative mode method was the same as described in the above paragraph. Positive mode method applied the same gradient and electrospray ionization parameters with eluent A consisting of 0.1% formic acid (v/v) in water and eluent B consisting of 0.1% formic acid (v/v) in methanol. Acquisition was done in *All-Ions* fragmentation mode using collision energies of 0, 20, and 40 eV. Feature identification was performed with MS-DIAL (version 4.24) by searching either (a) our own in-house authentic standard library[40] or (b) by comparison to MSMS spectra available on the Mass Databank of North America (MoNA). Search parameters included: mass accuracy: MS1 tolerance: 0.005 Da, MS2 tolerance: 0.025 Da. Peak detection minimum peak height: 2000 amplitude, mass slice width: 0.05 Da. Deconvolution parameters: sigma window value: 0.5, MS/MS abundance cut off: 5 amplitude. Identification MSP file: LC-MS/MS Spectra downloaded from MoNA database, accurate mass tolerance (MS1): 0.01 Da, accurate mass tolerance (MS2): 0.05 Da. Retention time and accurate mass based library: Stanford compound library, retention time tolerance: 0.1 min, accurate mass tolerance: 0.01 Da. Alignment parameters: reference file: QC01, retention time tolerance: 0.1 min, MS1 tolerance: 0.015 Da. Peaks from both positive and negative mode were normalized with the height of 2,2-$d_2$-hippuric acid, and

exported to MS-CleanR[41] for degenerate feature removal. Minimum blank ratio of 0.8, incorrect mass, and relative mass defect of 50–3000 were applied as filters. Pearson correlation ≥ 0.8 was used to compute clusters and the most intense peaks were kept in each cluster. Compounds annotated by retention time/accurate mass library were selected for targeted MS/MS spectra collection and compared with reference standards for confirmation of identity (all data provided in Supplementary Data 3).

### Analysis of additional metabolites of interest in MCAD[−/−] vs. MCAD[+/+] mice

The same samples on which untargeted metabolomics were performed were further analyzed for metabolites of interest identified during untargeted analyses (Supplementary Data 4). Quantitation of phenylpropionic acid, *N*-cinnamoylglycine, phenylpropionylglycine, 4-hydroxyhippuric acid, 3-(4-hydroxyphenyl)propionic acid, *N*-(3-(4-hydroxyphenyl)propionyl)glycine, and isocaproylglycine was performed using the same instrument and column under ESI negative mode. Eluent A consisted of 0.1% acetic acid (v/v) in water and eluent B consisted of 0.1% acetic acid (v/v) in methanol. Samples (0.5 µL) were injected via refrigerated autosampler into mobile phase and chromatographic separation at 50 °C was achieved at a flow rate of 0.3 mL min$^{-1}$ using the following gradient: 0–1 min, 1% B, 1–9 min, 1–99% B, 9–11 min, 99% B, 11–11.5 min, 99–1% B, 11.5–16 min, 1% B. Electrospray ionization parameters included fragmentor voltage of 70 V, gas temperature of 250 °C, sheath gas temperature of 250 °C, and capillary voltage of 4000 V. Peak areas were normalized using the internal standard and concentrations were determined by comparison to calibration curves prepared from dilution series of authentic standards. The calibration range for phenylpropionic acid was 0.020–30 mM, 0.003–12 mM for phenylpropionylglycine, 0.020–30 mM for 4-hydroxyhippuric acid, 0.049–30 mM for 3-(4-hydroxyphenyl)propionic acid, 0.020–12 mM for *N*-(3-(4-hydroxyphenyl)propionyl)glycine, 0.003–12 mM for isocaproylglycine, and 0.12–30 mM for *N*-cinnamoylglycine. Linear calibration regression model was applied to *N*-(3-(4-hydroxyphenyl)propionyl)glycine and creatinine, and quadratic regression model was applied to the other compounds, using $1/x$ weighed least-squared regression algorithm. $R^2 = 0.999$, 0.999, 0.999, 0.998, 0.998, 1.000, and 0.990 for creatinine, isocaproylglycine, 4-hydroxyhippuric acid, 3-(4-hydroxyphenyl)propionic acid, *N*-(3-(4-hydroxyphenyl)propionyl)glycine, phenylpropionylglycine, and *N*-cinnamoylglycine.

### Analysis of plasma acylcarnitine species by mass spectrometry

Acylcarnitine profiles were evaluated using a clinically validated isotope dilution mass spectrometry assay with a SCIEX 4500 triple quadrupole mass spectrometer. The full set of unlabeled (cat #s: NSK-B-US-1, NSK-B-G1-US-1, ULM-7195-PK) and labeled (cat #s: NSK-B, NSK-B-G1, DLM-9067-0.1MG) acylcarnitine standards were purchased from Cambridge Isotopes (Tewksbury, MA). For sample preparation, 20 µL of plasma were precipitated with 300 µL of acetonitrile with 0.1% formic acid containing the full set of isotopic internal standards. After centrifugation at $13,000 \times g$ for 5 min, the supernatants were transferred to a new tube and evaporated with nitrogen. Then, samples were derivatized with 100 µL of 3 N hydrochloric acid in butanol at 65 °C for 15 min. After derivatization, the samples were evaporated with nitrogen. Then, the dried samples were reconstituted with 80% acetonitrile in water and transferred to 96-well autosampler vials fitted with a silicone cap. The refrigerated autosampler was set to 4 °C. Samples (4 µL) were injected immediately after sample preparation. Extracts were introduced into the mass spectrometer by flow-injection analysis. The isocratic mobile phase conditions were 100% acetonitrile with a flow rate of 0.1 mL min$^{-1}$. Acylcarnitine species were detected in positive polarity ESI using a precursor ion scan of 85 $m/z$ over the mass range of 200–550 $m/z$. The electrospray ionization parameters were

5500 V for ionspray voltage, 200 °C for source temperature, and a pressure of 20 PSI for gases 1 and 2. For quantitation, peak heights were normalized using the internal standard, and concentrations were determined by comparison to calibration curves prepared for each acylcarnitine.

### Quantification and statistical analysis
MS-DIAL was used for peak-calling and integration of MCAD$^{+/+}$ vs. MCAD$^{-/-}$ LC-MS untargeted metabolomics data. All other analysis and visualizations were performed with Excel and GraphPad Prism (versions 8 and 9, graphpad.com).

### Reporting summary
Further information on research design is available in the Nature Portfolio Reporting Summary linked to this article.

## Data availability
The untargeted mass spectrometry data generated in this study have been deposited in the Metabolomics Workbench database. Raw GC-TOF shotgun metabolomics data is available under the accession number ST001606; untargeted LC-MS data, ST001633. Our analyses of these datasets are provided in Supplementary Data 1–4; all remaining data generated in this study are provided in the Source Data file. Source data are provided with this paper.

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

## Acknowledgements

We thank Lalla Fall for technical assistance with LC-MS analysis, Shuo Han for assistance with curation of authentic standards, Tim Meyer for provision of uremic solutes, and the ChEM-H Metabolomics Knowledge Center at Stanford University for assistance with LC-MS method development and access to instrumentation. This work was funded in part by a Ford Foundation Predoctoral Fellowship and National Science Foundation Graduate Research Fellowship Program (GRFP) to K.M.P., National Institutes of Health grants DK110335 (D.D.), GM142873 (D.D.), AT011396 (D.D.), DK101674 (J.L.S.), DK085025 (J.L.S.). J.L.S. and M.A.F. are Chan Zuckerberg Biohub Investigators.

## Author contributions

Statement D.D., J.L.S., M.A.F., and T.M.C. conceived and designed this study. K.M.P., D.D., W.V.T., J.B.J., Y.L., and S.H. performed mouse experiments and aided with sample preparation. H.C., D.D., J.M., B.W., and C.F. designed and performed LC-MS assays. All authors provided intellectual contributions. K.M.P., H.C., D.D., J.L.S., and M.A.F. wrote the paper, and all authors provided feedback.

## Competing interests

D.D. and M.A.F. are cofounders of Federation Bio. All remaining authors do not declare competing interests.
