## [Peer Review File · Nature Communications]

REVIEWERS' COMMENTS

Reviewer #1 (Remarks to the Author):

Pruss K.M., et al have studied the intersection between gut microbiota generated metabolites and host metabolism. Building on their previous work identifying the role of the *fld* locus in the metabolism of aromatic amino acids, they identified additional gut products subsequently metabolized by the host.

The authors have identified a clear objective – finding additional metabolites from the *fld* locus. The presented results advance our understanding of microbial-host interactions. This was a well written paper.

A major strength of the manuscript is that the authors have utilized a number of different complementary techniques such as shot-gun metabolomics, stable isotope labeling, generation of germ-free *MCAD*^{-/-} mice, and untargeted metabolomics dependent on *C. sporogenes* colonization.

The authors have built a complete ‘blueprint’ that could be used by other researchers to investigate or exploit microbial metabolite-host metabolism relationship in the context of numerous metabolic conditions.

Minor concerns

1) Attempting to validate their findings in conventional microbiota raised some surprising results. In the section «HA and PPA levels vary in conventional mice from different vendors», exploring more thoroughly the microbiota from mice from the different vendors would have been informative and valuable to the field. There is missed opportunity to compare the microbiota’s composition to see if indeed Jackson mice have HA producers, like *C. sporogenes* or other *fldC* gene containing microbes.

2) In figure 1 panel A: Adding the categories arylpropionates, N-acylglycines and arylacetates on the metabolic conversion pathway arrows would help better situate these categories of metabolites in aromatic AA catabolism. I appreciate the panel g but adding the categories would help the understanding.

3) In the same line of thought, adding a supra category with «fldC upstream metabolites», «fldC downstream metabolites» and «fldC alternative pathways» would help interpreting the heatmaps in panels D and F.

4) In the second paragraph of the results, I would add why hippuric acid is relevant for human health in the list of three reasons why HA is notable.

5) An interesting point that could be discussed is the molecular purpose of having a glycine conjugation step before releasing the metabolites studied in the plasma. It is clear that the 1 carbon shortening step happens to harvest the most energy out of the metabolites before excretion, but the glycine incorporation step seems conserved for all the metabolites and the importance of that step is not discussed.

Andre Marette, PhD

Reviewer #2 (Remarks to the Author):

The manuscript by Press KM et al describes a role for medium-chain acyl-CoA

29 dehydrogenase (MCAD) in host-microbe generation of hippuric acid building upon previous studies by Fischbach, Sonnenburg and Dodd labs. In this study, the authors demonstrate *C. sporogenes* reductive aromatic amino acid metabolism by fldC produces phenylacrylic acid by metabolomics and with stable isotope tracing defined the conversion of phenylalanine to hippuric acid. Interestingly, mice from different vendors generate variable levels of hippuric acid, which was abrogated in germ-free animals. The authors then hypothesized MCAD may be involved in beta-oxidation of phenylpropionic acid and then derived and evaluated germ-free MCAD^{-/-} mice. Indeed, germ-free MCAD^{-/-} mice showed higher levels of phenylpropionic acid by MS and stable isotope tracing, which was confirmed by in vitro studies. Finally, the authors show this host-microbe co-metabolism pathway may also be involved in other aromatic amino acid metabolites. Overall, the manuscript is well written and experimentally sound with sufficient quantitative analysis and supporting information. It is interesting that an abundant metabolite such as hippuric acid has not been associated a specific function in vivo. Seems like the authors would be well suited to re-evaluate this given their expertise and specific tools/reagents. Would be great if the authors could add to the discussion of their wt and MCAD^{-/-} mice colonized with *C. sporogenes* or other microbiota have been explored for resistance to pathogen and/or intestinal inflammation (DSS

model). I am pleased to recommend publication after this discussion or other potential functions are included.

Reviewer #3 (Remarks to the Author):

This paper describes a multi-layered microbiota-based study using GF, WT, fldC miceto to identify microbiota derived metabolites, and uncovers the mechanistic origin of hippuric acid via phenylalanine. The effort used untargeted metabolomics as well as stable isotopes to decipher the mechanism. While this paper does not represent the significance of the authors' recent Nature paper on IPA, nonetheless it offers useful insight into metabolites derived by gut microbiota. Given this overall impressive paper, the primary issue I have is the lack of supporting data confirming the identification of all of these molecules.

Specific comments:

- 1) The initial citations on the drug-like impact of microbiota should include PNAS 2009 (currently ref 21), as this effort was the original to observe this correlation.
- 2) the observations in Figures 1 and 2 are well done and quite striking for across the GF, WT and the fldC, across the many different content types.
- 3) The primary criticism of this paper is that there does not appear to be any data supporting the identifications of these molecules. This should be rectified.

REVIEWERS' COMMENTS

Reviewer #1 (Remarks to the Author):

Pruss K.M., et al have studied the intersection between gut microbiota generated metabolites and host metabolism. Building on their previous work identifying the role of the *fld* locus in the metabolism of aromatic amino acids, they identified additional gut products subsequently metabolized by the host.

The authors have identified a clear objective – finding additional metabolites from the *fld* locus. The presented results advance our understanding of microbial-host interactions. This was a well written paper.

A major strength of the manuscript is that the authors have utilized a number of different complementary techniques such as shot-gun metabolomics, stable isotope labeling, generation of germ-free MCAD^{-/-} mice, and untargeted metabolomics dependent on *C. sporogenes* colonization.

The authors have built a complete 'blueprint' that could be used by other researchers to investigate or exploit microbial metabolite-host metabolism relationship in the context of numerous metabolic conditions.

Minor concerns

1) Attempting to validate their findings in conventional microbiota raised some surprising results. In the section «HA and PPA levels vary in conventional mice from different vendors», exploring more thoroughly the microbiota from mice from the different vendors would have been informative and valuable to the field. There is missed opportunity to compare the microbiota's composition to see if indeed Jackson mice have HA producers, like *C. sporogenes* or other *fldC* gene containing microbes.

We appreciate the reviewer's positive comments about the paper and agree with their statement that the drastically differing levels of hippuric acid and phenylpropionic acid levels by microbial community composition were surprising and intriguing. We agree that comparison of community composition would be interesting, but to determine whether specific organisms harbored either the *fldC* gene or other homologous aromatic amino acid metabolic loci would require short-read metagenomic sequencing plus long-read metagenome-assembled genomes, and we believe these experiments are beyond the scope of this current study.

2) In figure 1 panel A: Adding the categories arylpropionates, N-acylglycines and arylacetates on the metabolic conversion pathway arrows would help better situate these categories of metabolites in aromatic AA catabolism. I appreciate the panel g but adding the categories would help the understanding.

We agree with the reviewer that further clarification in figure 1 panel A would aid the reader in understanding the metabolism of interest outlined later. We have updated Fig. 1A accordingly.

3) In the same line of thought, adding a supra category with «fIdC upstream metabolites», «fIdC downstream metabolites» and «fIdC alternative pathways» would help interpreting the heatmaps in panels D and F.

We have added further annotation as suggested by the reviewer to help clarify panels D and F.

4) In the second paragraph of the results, I would add why hippuric acid is relevant for human health in the list of three reasons why HA is notable.

We agree that emphasis on the relevance of hippuric acid to human health will be of strong interest to readers. We added the following statement just after the three reasons hippuric acid is relevant to human health: “Furthermore, abundance of the end product of oxidative phenylalanine metabolism, the cardiotoxic molecule phenylacetyl-glycine⁸, was higher in the urine of *fIdC*-colonized mice, indicating that reductive microbial metabolism of phenylalanine may represent a healthy alternative.”

5) An interesting point that could be discussed is the molecular purpose of having a glycine conjugation step before releasing the metabolites studied in the plasma. It is clear that the 1 carbon shortening step happens to harvest the most energy out of the metabolites before excretion, but the glycine incorporation step seems conserved for all the metabolites and the importance of that step is not discussed.

We thank the reviewer for this suggestion, which we believe will help the reader understand the scope of MCAD in phase I/II metabolism of microbial metabolites and have added a note on this to the discussion, see lines 244-251.

Andre Marette, PhD

Reviewer #2 (Remarks to the Author):

The manuscript by Press KM et al describes a role for medium-chain acyl-CoA 29 dehydrogenase (MCAD) in host-microbe generation of hippuric acid building upon previous studies by Fischbach, Sonnenburg and Dodd labs. In this study, the authors demonstrate *C. sporogenes* reductive aromatic amino acid metabolism by *fIdC* produces phenylacrylic acid by metabolomics and with stable isotope tracing defined the conversion of phenylalanine to hippuric acid. Interestingly, mice from different vendors generate variable levels of hippuric acid, which was abrogated in germ-free animals. The authors then hypothesized MCAD may be involved in beta-oxidation of phenylpropionic acid and then derived and evaluated germ-free MCAD^{-/-} mice. Indeed, germ-free MCAD^{-/-} mice showed higher levels of phenylpropionic acid by MS and

stable isotope tracing, which was confirmed by in vitro studies. Finally, the authors show this host-microbe co-metabolism pathway may also be involved in other aromatic amino acid metabolites. Overall, the manuscript is well written and experimentally sound with sufficient quantitative analysis and supporting information. It is interesting that an abundant metabolite such as hippuric acid has not been associated a specific function in vivo. Seems like the authors would be well suited to re-evaluate this given their expertise and specific tools/reagents. Would be great if the authors could add to the discussion of their wt and MCAD^{-/-} mice colonized with *C. sporogenes* or other microbiota have been explored for resistance to pathogen and/or intestinal inflammation (DSS model). I am pleased to recommend publication after this discussion or other potential functions are included.

We thank the reviewer for their comments. While hippuric acid has not been linked with any human diseases, the oxidative alternative end-product of phenylalanine metabolism by *C. sporogenes*, phenylacetyl-glutamine (or phenylacetyl-glycine in mice) has been identified as associated with cardiac disease. We believe aromatic amino acid metabolism to hippuric acid is thus an healthy alternative, as we stated in the discussion (lines 278-284), and which we have added to the Results section (lines 101-104).

Secondly, we agree that further phenotyping of mice deficient in MCAD (MCAD^{-/-}) warrants further study, especially with respect to colonization states with different microbial consortia. However, while these studies are ongoing in our laboratory, we believe they are beyond the scope of the current study.

Reviewer #3 (Remarks to the Author):

This paper describes a multi-layered microbiota-based study using GF, WT, fldC miceto to identify microbiota derived metabolites, and uncovers the mechanistic origin of hippuric acid via phenylalanine. The effort used untargeted metabolomics as well as stable isotopes to decipher the mechanism. While this paper does not represent the significance of the authors' recent Nature paper on IPA, nonetheless it offers useful insight into metabolites derived by gut microbiota. Given this overall impressive paper, the primary issue I have is the lack of supporting data confirming the identification of all of these molecules.

Specific comments:

1) The initial citations on the drug-like impact of microbiota should include PNAS 2009 (currently ref 21), as this effort was the original to observe this correlation.

We thank the reviewer for pointing this out and we have included Wikoff *et al.* 2009 in our initial citations of host metabolism of molecules produced by the gut microbiota.

2) the observations in Figures 1 and 2 are well done and quite striking for across the GF, WT and the fldC, across the many different content types.

We thank the reviewer for this comment.

3) The primary criticism of this paper is that there does not appear to be any data supporting the identifications of these molecules. This should be rectified.

We apologize for not clearly presenting how the identifications were performed. We have followed the current best practices for metabolite identification in untargeted metabolomics which require accurate mass plus two orthogonal properties (e.g., MS/MS and retention time) determined from an authentic reference standard for the highest confidence structural assignment (Level 1) (ref here). We verified these annotation levels for all analytes reported in our manuscript as described in the methods section (lines 540-556) and all data is provided in supplemental table 3. To make this more clear to the reader, we included the following statement in the methods (lines 540-542): “Feature identification was performed with MS-DIAL (version 4.24) by searching either a) our own in-house authentic standard library (Han and van Treuren et al PMID: 34262212) or b) by comparison to MSMS spectra available on the Mass Databank of North America (MoNA).”